# A Novel Muscle Synergy Extraction Method Used for Motor Function Evaluation of Stroke Patients: A Pilot Study

**DOI:** 10.3390/s21113833

**Published:** 2021-06-01

**Authors:** Yehao Ma, Changcheng Shi, Jialin Xu, Sijia Ye, Huilin Zhou, Guokun Zuo

**Affiliations:** 1Cixi Institute of Biomedical Engineering, Ningbo Institute of Materials Technology and Engineering, Chinese Academy of Sciences, Ningbo 315300, China; mayehao@nimte.ac.cn (Y.M.); xujialin@nimte.ac.cn (J.X.); yesijia258@163.com (S.Y.); zhouhuilin@nimte.ac.cn (H.Z.); moonstone@nimte.ac.cn (G.Z.); 2Zhejiang Engineering Research Center for Biomedical Materials, Ningbo Institute of Materials Technology and Engineering, Chinese Academy of Sciences, Ningbo 315300, China

**Keywords:** muscle synergy, MCR-ALS, sparseness, electromyography, motor function, stroke

## Abstract

In this paper, we present a novel muscle synergy extraction method based on multivariate curve resolution–alternating least squares (MCR-ALS) to overcome the limitation of the nonnegative matrix factorization (NMF) method for extracting non-sparse muscle synergy, and we study its potential application for evaluating motor function of stroke survivors. Nonnegative matrix factorization (NMF) is the most widely used method for muscle synergy extraction. However, NMF is susceptible to components’ sparseness and usually provides inferior reliability, which significantly limits the promotion of muscle synergy. In this study, MCR-ALS was employed to extract muscle synergy from electromyography (EMG) data. Its performance was compared with two other matrix factorization algorithms, NMF and self-modeling mixture analysis (SMMA). Simulated data sets were utilized to explore the influences of the sparseness and noise on the extracted synergies. As a result, the synergies estimated by MCR-ALS were the most similar to true synergies as compared with SMMA and NMF. MCR-ALS was used to analyze the muscle synergy characteristics of upper limb movements performed by healthy (n = 11) and stroke (n = 5) subjects. The repeatability and intra-subject consistency were used to evaluate the performance of MCR-ALS. As a result, MCR-ALS provided much higher repeatability and intra-subject consistency as compared with NMF, which were important for the reliability of the motor function evaluation. The stroke subjects had lower intra-subject consistency and seemingly had more synergies as compared with the healthy subjects. Thus, MCR-ALS is a promising muscle synergy analysis method for motor function evaluation of stroke patients.

## 1. Introduction

How the central nervous system (CNS) controls the musculoskeletal system to solve the redundancy problem of degree of freedom (DOF) is an important research topic. One strategy widely recognized by a significant number of scholars is that the CNS accomplishes a variety of behaviors through statistical regularities involving biomechanical properties of the human body, and then synergistically applies these regularities to perform different motor tasks [1,2,3]. These regularities are called “muscle synergies”. The limb movements are accomplished by activating these synergies coordinately [3,4].

Although the physiological origin and meaning of muscle synergies are still debated [5], it has been confirmed that motor task execution can be described by the coordination of a limited number of muscle synergies. A muscle synergy represents the relative activation strengths of a group of muscles, simplifying the control of the musculoskeletal system. Muscle synergy provides a new method for studying the motor control mechanism during a movement [6]. Ivanenko et al. found five muscle synergies accounted for muscle activity during human locomotion [7]. Scano et al. proved that a large variety of grasps can be produced by a limited subset of muscle synergies [8]. Aoi et al. developed a motor model with 69 parameters based on muscle synergy, which produced both walking and running of a human musculoskeletal model by changing only seven motor control parameters and concluded a human could change walking and running speed through seven key motor control parameters [9]. Pan et al. found that the similarity of muscle synergies of subacute stroke survivors was significantly correlated with the Brunnstrom stage [10]. Cheung et al. considered muscle synergy to be physiological markers of motor cortical damage, and the muscle synergies of stroke survivors had three distinct patterns, i.e., preservation, merging, and fractionation [11]. Thus, a growing number of studies have been focused on the synergy characteristics of subjects with nervous system diseases, such as stroke [11,12,13], spinal cord injury [14,15], cerebral palsy [16], and Parkinson’s disease [17]. The results have shown that muscle synergy is a promising approach for motor function evaluation.

Various matrix decomposition algorithms have been applied to extract muscle synergy from recorded and processed electromyographic (EMG) signals of dynamic motor tasks. Nonnegative matrix factorization (NMF) [11,12,13,14,15,16,17,18,19,20,21], factor analysis (FA) [22,23], independent component analysis (ICA) [24,25], and principal component analysis (PCA) [26,27] are the four common synergy extraction algorithms. PCA produces the basis vectors (muscle synergies) with the best variance description of EMG data through singular value decomposition (SVD) [6]. Similar to PCA, FA extracts muscle synergy weights by calculating the eigenvectors of the data’s covariance matrix; the eigenvectors with eigenvalues >1 are considered to be the muscle synergies. The synergies are determined by employing singular value decomposition for analyzing the data’s covariance matrix. ICA makes the data statistically independent by transforming it orthogonally. ICA is employed to analyze the data with non-Gaussian variation and extract synergies that maximize the absolute value of the fourth moment of the data. In contrast to the aforementioned methods, NMF is the most widely used algorithm because of its non-negativity constraint and simple principle [6,15,16,17,18,19,20,21,28]. However, NMF has some limitations. For example, it is based on maximizing Gaussian likelihood, and easily falls into local optimum for identifying dependent and non-sparse components. Thus, some modified methods have been proposed. Several algorithms based on gamma and inverse Gaussian models have been used to analyze EMG data in order to deal with the influence of signal noise on muscle synergy extraction and have provided better reconstruction quality as compared with NMF [29,30]. The sparse NMF (SNMF) has been proposed to improve the performance of NMF for identifying dependent components [31]. In addition, constrained Tucker decomposition (consTD) has been employed to deal with the muscle synergy extraction problem from EMG data of various biomechanically related tasks [32]. However, there are few studies that have focused on non-sparse synergies extraction [33].

Some human muscles are biarticular and polyarticular muscles, which contribute to more biomechanical subtasks [31]. As a result, many synergies present a high degree of coactivation in certain movements with multi-joint or multi-degree of freedom. In addition, the abnormal coactivations of some muscles may increase the non-sparseness of synergies and activations for some disease patients, such as stroke [34]. It is well known that NMF presents excellent performance for extracting sparse components [35,36]. However, the performance of NMF may decrease in non-sparse synergy extraction, which significantly restricts the application of muscle synergy for evaluating motor function of stroke patients. To deal with the problem, a method associated with the characteristics of synergy mixture system itself is a feasible solution for non-sparse synergies extraction.

Self-modeling mixture analysis (SMMA) is a common signal resolution algorithm for a linear mixture system, and it can resolve component information by analyzing the statistical characteristics of various signal variables. Thus, as compared with the randomly initialized synergy matrix calculated by NMF, the synergy matrix produced by SMMA is much more consistent with the real ones. In this study, a SMMA-based synergy extraction approach called multivariate curve resolution–alternating least squares (MCR-ALS) was proposed and its performance was compared with SMMA and NMF. We evaluated the feasibility of the proposed algorithm with simulation data, for which the properties of the synergies and activations were known. The effects of the sparseness and noise intensity on the extracted synergies were investigated. In addition, the proposed method was used to analyze the muscle synergy of upper limb movements of stroke subjects; the intra-subject consistency and the number of muscle synergies between stroke and healthy subjects were compared and investigated.

## 2. Theory and Experiment

### 2.1. Muscle Synergy Pattern Model

According to the theoretical background of muscle synergy, EMG signals are considered to be the weighted summation of primitive functions reflecting the activation time information of several muscle groups, and their synergies (muscle groups) reflect the relative activation strengths of multiple muscles. Thus, muscle synergies can be extracted through unmixing EMG data as the following bilinear model [11,37]:(1)D=CS+E
where *D* is the *m* × *n* matrix which consists of a set of *m* preprocessed EMG signals and *n* data points for each EMG signal, correspondingly; *C* is the *m* × *r* matrix of weight coefficient describing the enrollment of muscle groups (muscle synergy); *r* represents the number of muscle synergies; *S* is a *r* × *n* matrix of activations constituting the basic components or primitive functions; and *E* is residual matrix with size of *m* × *n*. Every muscle synergy is a time-invariant module reflecting relative activation strengths of multiple muscles, which is activated by time-varying commands (activation) descending from CNS [11].

### 2.2. Simulated Data

A simulation study is a useful approach for synergy extraction algorithm assessment as the properties of the synergies and activations are known. In this study, various simulated data sets with different noise intensities were constructed to evaluate the accuracy and robustness of the proposed synergy extraction algorithm. According to the muscle synergy pattern model (Equation (1)), simulated data, *D_sim_*, was generated according to following equation:(2)$$D_{sim}=C_{sim}S_{sim}+E_{sim}$$
where *D_sim_* is a 10 × 1000 matrix which consists of a set of 10 simulated EMG signals (muscles or channels) and 1000 data points for each EMG signal, correspondingly; *C_sim_* is a 10 × 4 weight matrix containing 4 synergies; *S_sim_* is a 4 × 1000 matrix of activation; and *E_sim_* is the signal-dependent noise matrix with a size of 10 × 1000.

In past studies, researchers have proven that the noise in neural control signal increases with an increase in the magnitude of signal [38]. Thus, in this study, the noise, whose standard deviation (SD) is positive proportional to noiseless signal value, was added to the EMG signal [39]. The scale factor *a* was used to change the noise intensities of EMG signals; its value varied from 0.05 to 0.15. The signal-dependent noises with *a* = 0.05, 0.10, 0.15 (SNR = 26, 20, 16 dB) were added to noiseless signal to generate different simulated data sets, according to Equation (2).

In this study, to evaluate the effect of the sparseness on the proposed synergy extraction method, the synergies and activation profiles with six degrees of sparseness (0.1, 0.2, 0.3, 0.4, 0.5, 0.6) were used to create the simulated data. The synergies with various degrees of sparseness were created randomly and their values were between [0, 1]. The activation profiles with various degrees of sparseness were generated by linear combination of Gaussian and Lorentz functions by control of the full width at half maximum. The degree of sparseness of vector *x* (synergy or activation profile) was evaluated by Equation (3) [40], which is:(3)$$sparseness(x)=\frac{\sqrt{n}-(\sum \left|x_{i}\right|)/\sqrt{{\sum x}_{i}^{2}}}{\sqrt{n}-1}$$
where *n* is the dimension of *x*, and *x_i_* is the *i*th variable of *x*. In the following paragraphs, the sparseness of synergies and activations represents the average sparseness of all synergies and all activations in muscle synergy pattern model, respectively.

### 2.3. Experimental Data

Eleven healthy subjects (age range 25–37 years, mean age 26.7 years) and five stroke subjects (Brunnstrom 3–5, upper limb Fugl-Meyer score of 22–60, arm Fugl-Meyer score of 16–36, age range 52–79 years, mean age 67.8 years) volunteered to participate in this study. The healthy subjects had no neurological disorders and limb surgeries in their clinical histories. The stroke subjects had no limb surgeries in their clinical histories. The subjects were asked not to drink alcohol and caffeine 24 h before the experiment. The experiment was performed based on the Declaration of Helsinki and confirmed by the Ethics Committee of Cixi Institute of Biomedical Engineering. 

The EMG data used in this study were collected from upper limb movements based on an upper limb rehabilitation robot. Subjects sat on a chair with their back straight and perpendicular to the ground. The robot resistance was adjusted to a uniform and comfortable level for each subject. Each subject performed upper limb movements according to the trajectory (semicircle) on the screen of the rehabilitation robot, shown in Figure 1. In the experiment, subjects sat on the side of the upper limb rehabilitation robot and operated a shot stick using their evaluated upper limb (dominant hand for healthy subject and affected hand for patients) to accomplish the task repeatedly (six trials for each subject). To improve the repeatability of the experiment, the motion range of the shot stick was limited by a computer program that ensured it moved along a semicircle trajectory. In any trial, each subject put his hand on the robot’s tray and grasped the shot stick with five fingers on the starting position. The experimenter started the data acquisition and give the “go” signal. The subjects did the movement with the shot stick after the “go” signal and stopped at the ending position shown on the screen. The consumed time of the movement was about 2 s. Data collection stopped automatically after 3 s. 

In this study, the activities of seven muscles, including the anterior deltoid (DA), posterior deltoid (DP), triceps brachii (TI), biceps brachii (BI), extensor carpi radialis (ECR), flexor carpi radialis (FCR), and brachioradialis (BIO), were recorded through a 16-channel EMG system (Delsys Inc., Boston, MA, USA). Before placing the EMG sensors, excessive hair was shaved from the skin and alcohol was applied to wipe the skin to remove oils and surface residues. Each EMG sensor was placed on the muscle belly under the guidance of a therapist. The raw EMG signals were recorded with a sampling frequency of 1926 Hz. Figure 2 presents the raw EMG signals of seven muscles recorded during a trial of one healthy subject. To attenuate DC offset and high-frequency noise, the raw EMG signals were band-pass filtered between 40 and 400 Hz (3th order zero-lag Butterworth) [41]. The filtered EMG signals were full-wave rectified and low-pass filtered at 5 Hz (3th order zero-lag Butterworth) to calculate the envelopes of EMG signals [41,42]. The envelopes normalized by the maximum of each envelope itself were used to extract muscle synergy [12]. 

## 3. Methodology

### 3.1. NMF

NMF, which was first proposed by Lee and Seung [35,36], is usually used to reduce data dimension. Because of its nonnegative constraint, NMF is also applied to analyze medical [43] and space object [44] data. 

NMF is the most widely used muscle synergy extraction method [15,16,17,18,19,20,21], which is usually based on the multiplicative update rules [36]. After creation of the random initial matrices (*C* and *S*), the iteration is to minimize the Frobenius norm of the residual matrix (preprocessed EMG matrix *D* minus multiplication of the matrix *S* and *C*) illustrated by Equation (4). The stop criterion is set based on the parameter Q, the percentage of change in the lack of fit between two iterations (*f_l_*(*S*,*C*) and *f_l_*_+1_(*S*,*C*)), which is obtained through Equation (5). The stop criterion is: (1) Q equal to 0.01% and (2) max number of iterations equal to 1000. Equations (4) and (5) are as follows:(4)$$f(S,C)=\frac{1}{2}{\left\|D-CS\right\|}_{F}^{2}$$
(5)$$Q=100\cdot \left(\frac{f_{l+1}(S,C)-f_{l}(S,C)}{f_{l}(S,C)}\right)$$

### 3.2. SMMA

SMMA, which is also called simple-to-use interactive self-modeling mixture analysis (SIMPLISMA), is a matrix decomposition method for a linear mixture system. SMMA is proposed based on pure variables, which contribute from one component [45]. Thus, SMMA resolves the signal of mixtures by analyzing the characteristics of various variables of signal, and then extracts the values of the variables containing quantitative information to resolve the linear mixture system. SMMA is usually used in a signal analysis of a chemical mixture system [46]. According to Equation (1), muscle synergy is also a linear mixture system and the EMG signals are considered to be weighted summation of time-variant activation. The nonstationarity of the activation profile demonstrates that it is feasible to find pure variables in EMG data. The pure variables, which contain the relative content information of components (synergies), are used to resolve the components (activations).

### 3.3. MCR-ALS

MCR-ALS, which evolved from SMMA, is a popular matrix factorization algorithm [47,48]. For MCR-ALS, the initial synergy and activation matrices, containing relative content and component information, respectively, are obtained from SMMA. In addition, alternating least squares (ALS) is used to optimize the resolution according to Equations (6) and (7) [49], and nonnegative constraint is imposed to ensure that the decomposition results have physical significance. In the iterative process, if the elements of *S* and *C* are negative, it is set to zero forcibly. The stop criterion of MCR-ALS is the same as NMF (Q equal to 0.01%, max number of iteration equal to 1000). Equations (6) and (7) are as follows:(6)$$C=(DS^{T}){\left(SS^{T}\right)}^{-1}$$
(7)$$S={\left(C^{T}C\right)}^{-1}C^{T}D$$

### 3.4. Algorithm Evaluation

To compare the performances of the three algorithms, they were applied to analyze the simulated data created by synergies and activations with different sparseness and noise levels. The Pearson’s correlation coefficient between true and extracted synergies was applied to assess the three muscle synergy extraction algorithms. The match between extracted and true synergies was performed based on their similarity evaluated by the Pearson’s correlation coefficient. We paired the true and extracted synergies with highest similarity. If more than one extracted synergy had the greatest similarity to the same true synergies, simultaneously, we achieved full math by studying the total correlation of all synergy pairs. The match pattern with the max sum of correlation coefficient was used as the final match result. The statistical results (performances on 25 simulated data containing randomly created synergies and activations for each sparseness combination and noise level) of the correlation coefficients of full matched synergies were used to assess the accuracy and robustness of different synergy extraction methods.

In the experimental study, the repeatability of different algorithms was evaluated by analyzing the variation of repeated estimated synergies (25 times) utilizing Pearson’s correlation coefficient. The evaluation was based on the Pearson’s correlation coefficient between matched synergies extracted from any two performances. Similar to the simulation study, the first step was to match the synergies extracted from two calculations for the same data set (trial). The pair of synergies with the highest correlation were matched together. If two or more synergies had the maximum correlation coefficient with the same synergy, a full match was realized by analyzing the overall correlation of all synergy pairs. The match pattern with the max sum of correlation coefficient was considered to be the final match result. The correlation coefficients of all possible combinations of full matched synergies extracted from 25 repeated performances were used to assess the repeatability of the three synergy extraction algorithms.

In this study, the intra-subject consistency of synergies across multiple trials of each subject was applied to assess the robustness of the proposed synergy extraction algorithm. The intra-subject consistency was calculated by analyzing Pearson’s correlation coefficient between each pair of trials for each subject. For each pair of trials, the correlation coefficients of all possible combinations of full matched synergies from the two trial data sets, respectively, were calculated. For each subject, the average of the correlation coefficients of all trial combinations was used to represent intra-subject consistency. A paired *t*-test was used to analyze the difference in intra-subject consistency for various algorithms. One-way ANOVA was performed to detect the difference in intra-subject consistencies between healthy and stroke subjects.

### 3.5. Choose the Number of Synergies

In the analysis described above, we assumed that the correct number of synergies was known before decomposition. However, in many situations, the number of synergies, *r*, is unknown. In this study, we applied variance account for (VAF) to determine the number of muscle synergies *r* [50,51,52]. VAF was calculated according to the following equation:(8)$$VAF=1-({\left\|D-M\right\|}^{2}/{\left\|D-mean(D)\right\|}^{2})$$
where *M* = *CS*, which represents the reconstruction matrix of synergy extraction algorithm; the operator “mean” is to produce a matrix with the same size as *D*, whose columns are made up of the mean of corresponding column in *D*. Here, *r* is defined as the minimum when VAF exceeded 80% [11,13].

## 4. Results

### 4.1. Evaluation with Simulated Data

In order to evaluate the feasibilities of the three muscle synergy extraction algorithms, we employed various simulated EMG datasets generated from components (synergies and activation profiles) with different sparseness combinations. The similarity between estimated and true components was used to assess various synergy extraction methods. As a result, the performance of NMF was affected by both the sparseness of synergies and the sparseness of activations. NMF became less accurate as the degree of sparseness decreased. In contrast, SMMA and MCR-ALS were mainly affected by activation sparseness, because the purities of pure variables were easily influenced by the non-sparseness of activations. However, the influence of the non-sparseness of synergies on the two methods was relatively small. Figure 3 presents the performances of NMF, SMMA, and MCR-ALS versus synergy sparseness and activation sparseness under the circumstance of noise level 0.05. Obviously, the performance of NMF was different from SMMA and MCR-ALS.

To validate the robustness of each synergy extraction method, we analyzed their performances under circumstances of different noise intensities. Figure 4 shows the average correlation coefficients for the fully identified synergies compared across two settings (sparseness, noise) for the three methods. From the figure, it can be seen that the noise was an important influence factor for synergy extraction. The estimated synergies became less accurate as the noise level increased. In the three methods, MCR-ALS provided the best performance, especially when sparseness of synergy and activation was low. For example, the average correlation coefficient of synergy estimated by MCR-ALS was 0.96 when the synergy sparseness and noise level were 0.1 and 0.05, while the average correlation coefficients of synergies estimated by SMMA and NMF were 0.92 and 0.83, respectively. As the synergy sparseness increased, the performances of NMF and MCR-ALS became close. Lacking iterative optimization restricted the performance of SMMA. In addition, the non-negativity of its decomposing results was hard to be ensured. For the data containing non-sparse components (synergy or activation), NMF easily fell into local optimum and its result was non-unique, which lead to its poor performance. However, MCR-ALS could supply accurate and unique decomposing results.

### 4.2. Results of Motor Function Evaluation by Muscle Synergy

In the simulation study, SMMA provided unique decomposition as compared with NMF, but it still could not present satisfactory performance. MCR-ALS, a developed algorithm which evolved from SMMA, optimized the resolution through ALS iteration. Thus, MCR-ALS was the most reliable and accurate synergy extraction. In this study, MCR-ALS was also applied to analyze the muscle synergies of upper limb movements of stroke subjects, to improve the performance of muscle synergy on motor function evaluation.

For each healthy subject, three muscle synergies were needed to ensure the reconstruction accuracy for the EMG data. For the conventional method (NMF), muscle synergies were extracted with an average VAF value of 85.67 ± 5.64%. However, for the proposed method (MCR-ALS), the muscle synergies were identified with an average VAF value of 86.32 ± 4.97%. The number of synergies of stroke subjects was different from healthy subjects. Three muscle synergies were enough to reconstruct the EMG data accurately for three stroke subjects, while for the other two stroke subjects, four muscle synergies were needed to realize accurate data reconstruction. Muscle synergies were extracted by NMF with an average VAF value of 86.35 ± 5.44%. For MCR-ALS, the muscle synergies were identified with an average VAF value of 86.47 ± 5.39%. Obviously, the two methods have close reconstruction ability. 

To compare the repeatability of MCR-ALS and NMF, we analyzed 25 repeated decomposition results for each trial dataset. As a result, the average repeatability of NMF was 0.86 ± 0.11 and 0.74 ± 0.13 for all healthy and stroke subjects, respectively. Figure 5 is the boxplot of the correlation coefficients of muscle synergies extracted from a typical trial EMG data of healthy subject. As seen in Figure 5, the average correlation coefficients of three synergies were 0.92, 0.80, and 0.65, respectively, with box lengths of 0.08, 0.25, and 0.52. Obviously, NMF presented inferior repeatability for the upper limb movement data, and it usually fell into local optimum. However, MCR-ALS could extract relative concentration information through pure variables in the initialization phase instead of random initialization, thus, its repeatability was one. In other words, the initialized synergy and activation matrix is unique.

Figure 6 shows the muscle synergies extracted by NMF and MCR-ALS from one healthy subject. As seen in the figure, the synergies produced by NMF had much larger standard deviations as compared with the ones identified by MCR-ALS, especially for Synergy 2 and Synergy 3. As shown in Figure 6, the synergies extracted by NMF and MCR-ALS all had low sparseness, especially Synergy 2 and Synergy 3. Thus, the performance of NMF might be affected by the non-sparseness of synergy. A similar conclusion can be obtained through the synergy analysis of stroke subjects. As Figure 7 shows, the standard deviations of the synergies extracted by MCR-ALS were much smaller than the ones identified by NMF.

To further evaluate the performance of the two methods, intra-subject consistency was calculated. Figure 8 represents the statistical results of intra-subject consistency computed to compare the standard approach (NMF) and the novel approach (MCR-ALS) for healthy and stroke subjects. For the healthy subjects, there was a significant difference in intra-subject consistency between the two methods (*p* < 0.001). The consistencies of MCR-ALS and NMF were equal to 0.86 ± 0.04 and 0.69 ± 0.09, respectively. For the stroke subjects, there was also a significant difference in intra-subject consistency between the two methods (*p* = 0.01). The consistencies of MCR-ALS and NMF were equal to 0.56 ± 0.11 and 0.38 ± 0.15, respectively. Obviously, NMF had inferior consistency as compared with MCR-ALS. The main reason may be that NMF could not obtain optimum in non-sparseness component extraction. From the experimental study, MCR-ALS could provide more reliable synergy identification as compared with NMF.

Compared with healthy subjects, some chronic stroke subjects had more synergies, which was consistent with those reported in the literature [11]. The two groups of subjects had a significant difference in intra-subject consistency (F = 78.275, *p* < 0.001 for MCR-ALS and F = 27.835, *p* < 0.001 for NMF). Stroke subjects had inferior synergy consistencies as compared with healthy subjects (0.38 ± 0.15 vs. 0.69 ± 0.09 for NMF and 0.56 ± 0.11 vs. 0.86 ± 0.04 for MCR-ALS), which demonstrated that stroke patients had poorer motor control ability as compared with healthy subject. Thus, the number of synergies and intra-subject consistency are promising indexes used for motor function evaluation for stroke patients.

## 5. Discussion

In the simulation study, NMF showed good performances for identifying sparse synergies. However, for non-sparse synergy extraction, NMF could not provide satisfactory performance. The estimated synergies had a large error as compared with true synergies. Many human muscles are biarticular and polyarticular, which induces the phenomenon of coactivation and non-sparse synergies in many human movements with multiple degrees of freedom. The experiment of this study is a great example, the synergies of upper limb movements involving digital, wrist, elbow, and shoulder joints are non-sparse. In addition, the abnormal coactivations of stroke patients’ muscles also affects the sparseness of synergies. The analysis results of the experimental data showed that NMF easily fell into local optimum in extracting non-sparse components.

From the results of the simulation study, SMMA was a feasible method for identifying muscle synergy. In addition, its decomposing result was unique. However, the non-sparse activations affected the purities of pure variables. In other words, it was hard to find several data points completely contributed from one synergy. Thus, SMMA could not provide satisfying resolution. However, MCR-ALS, as a developed algorithm, optimized the purities of pure variables through ALS. As a result, MCR-ALS was the most reliable synergy identification method, even though the EMG data contained non-sparse synergies and non-sparse activations simultaneously.

Predicting the number of synergies is an important task for muscle synergy analysis. We applied VAF to predict the number of synergy vectors ensuring enough reconstruction accuracy of the matrix decomposition algorithm for EMG data. In the analysis of the EMG study (healthy and stroke subjects), the VAF values of NMF and MCR-ALS were very close. Thus, MCR-ALS also had enough learning ability for EMG data, ensuring a robust prediction of the number of synergies.

The non-negativity is also a key index for the matrix decomposition algorithm. The SMMA usually produces negative values resulting from the amplitude differences between the pure variables and weight of synergies. Thus, non-negative constraint is necessary for SMMA to ensure the physical significance of decomposition. However, the relative relationship between pure variables and weight of synergies would remain if the non-sparseness of activation profiles were slight. Thus, the synergies were still similar to real ones. MCR-ALS evolved from SMMA, and its non-negative constraint ensures the positive decomposing results. As compared with the pure mathematical method (NMF), MCR-ALS and SMMA are strongly associated with system theory and the structure of the data itself, thus, they have greater robustness. 

For stroke patients, their synergies have a low degree of sparseness because of the abnormal muscle activations and movements with multiple degrees of freedom. The poor reliability of NMF severely restricts the application of muscle synergy to motor function evaluation. MCR-ALS supplied more reliable muscle synergy information as compared with NMF, contributing to the robust motor function evaluation. In addition, chronic stoke subjects might have more synergies and inferior intra-subject consistency as compared with healthy subjects. Therefore, the two parameters are two promising indexes used for motor function evaluation of stroke survivors. In future study, we plan to analyze the muscle synergies of more stroke subjects with the novel EMG analysis method.

## 6. Conclusions

In this study, a novel muscle synergy extraction method called MCR-ALS was proposed. Its performance was compared with two other matrix decomposition algorithms (NMF and SMMA). The results showed the following: (1) The problem of non-unique decomposition of NMF was resolved through pure variable extraction (SMMA and MCR-ALS); (2) As a developed algorithm evolved from SMMA, MCR-ALS presented the greatest reliability in synergy identification as compared with NMF and SMMA, especially for the data containing non-sparse components. In addition, MCR-ALS was used to analyze the muscle synergy characteristics of stroke subjects. Through comparative study, stoke subjects have more synergies and inferior intra-subject consistency as compared with healthy subjects. Therefore, MCR-ALS is a promising muscle synergy extraction method. The results of this study are of great significance for promoting the application of muscle synergy for evaluating motor function of stroke patients.

## Figures and Tables

**Figure 1 sensors-21-03833-f001:**
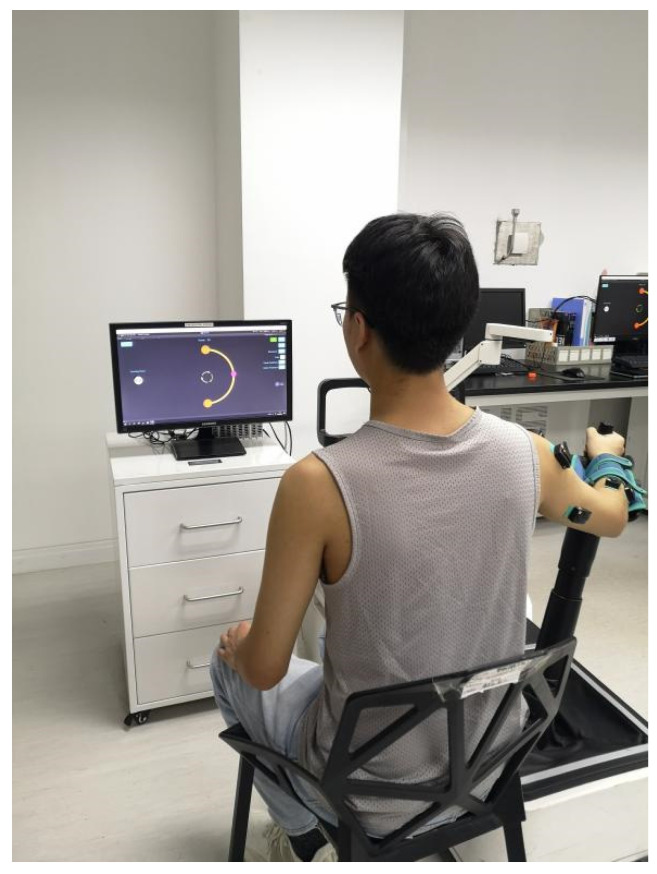
The upper limb movement experiment based on the rehabilitation robot.

**Figure 2 sensors-21-03833-f002:**
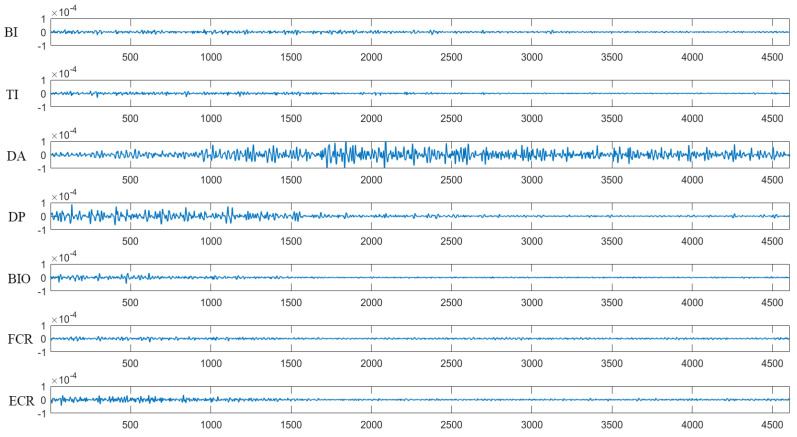
Raw EMG data recorded during the trial of one healthy subject. The unit of the Y coordinate is V.

**Figure 3 sensors-21-03833-f003:**
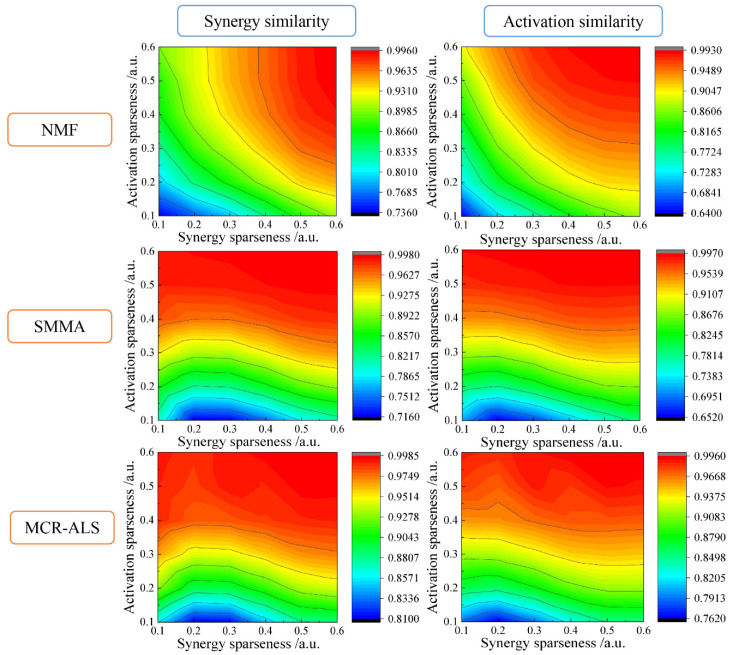
The performances of NMF, SMMA, and MCR-ALS versus synergy sparseness and activation sparseness under the circumstance of noise level 0.05. (The color represents the degree of similarity evaluated by the Pearson’s correlation coefficient between the synergies and activations extracted by the matrix decomposition algorithms and the true synergies and activations; the isolines represent the same value of average correlation coefficients; a.u. represents arbitrary unit).

**Figure 4 sensors-21-03833-f004:**
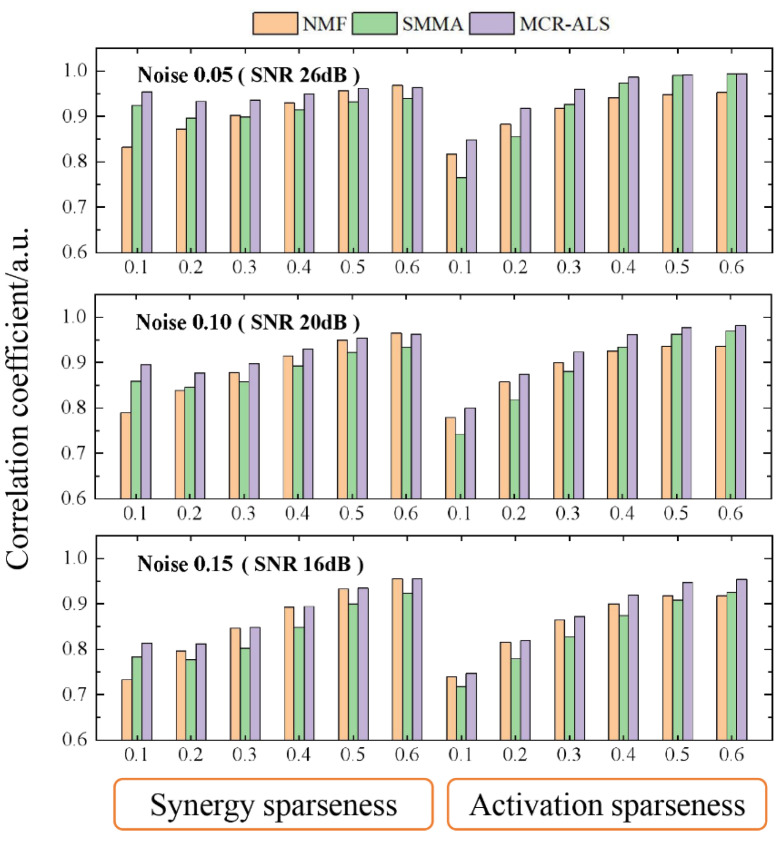
Average correlation coefficients for the fully identified synergies compared across two settings (sparseness, noise) for the three methods.

**Figure 5 sensors-21-03833-f005:**
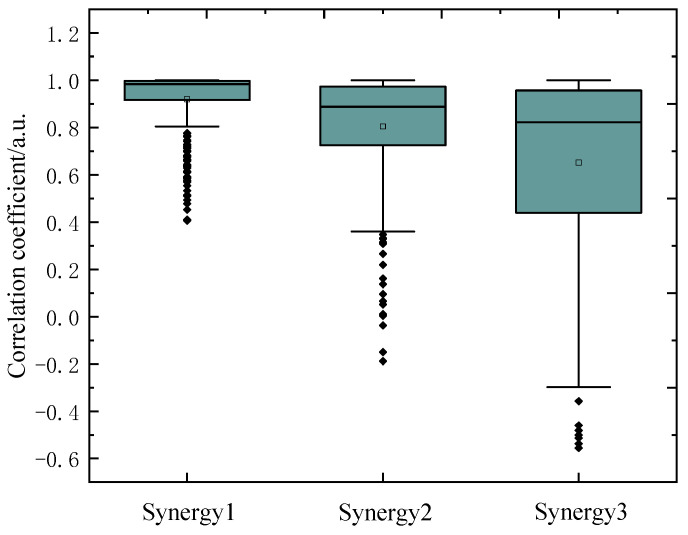
The boxplots of the correlation coefficients of muscle synergies extracted from a typical trial data of healthy subject using NMF.

**Figure 6 sensors-21-03833-f006:**
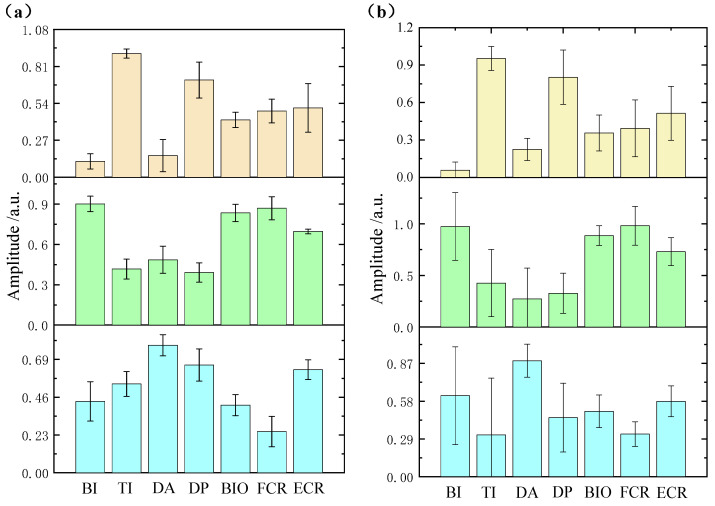
The statistical chart of synergies extracted by MCR-ALS (**a**) and NMF (**b**) from a typical healthy subject.

**Figure 7 sensors-21-03833-f007:**
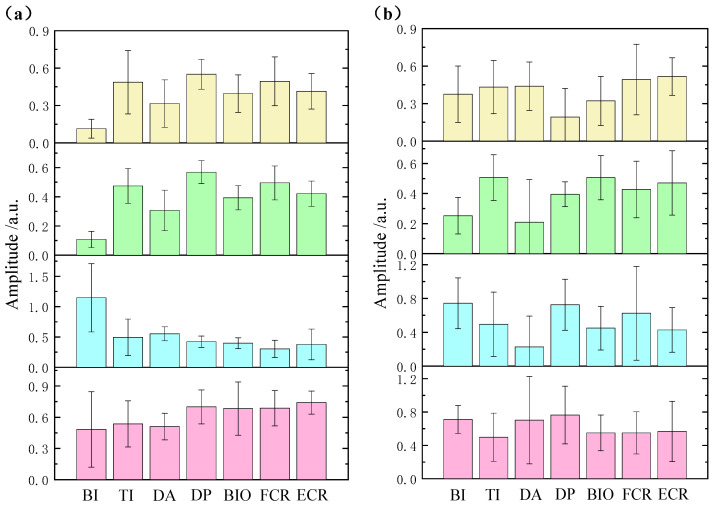
The statistical chart of synergies extracted by MCR-ALS (**a**) and NMF (**b**) from a typical stroke subject.

**Figure 8 sensors-21-03833-f008:**
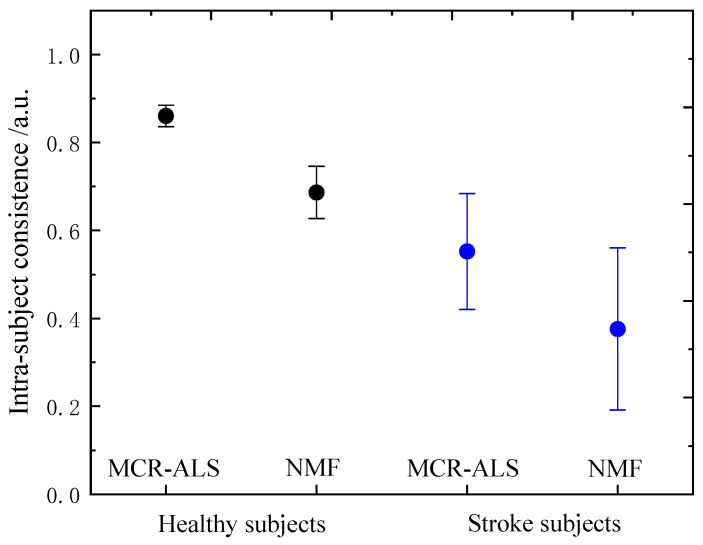
The statistical results of intra-subject consistency computed to compare the standard approach (NMF) and the novel approach (MCR-ALS) for all healthy and stroke subjects.

## Data Availability

The data presented in this study are available on request from the corresponding author.

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
