# Peer review of "A Novel Muscle Synergy Extraction Method Used for Motor Function Evaluation of Stroke Patients: A Pilot Study"

_sensors, 2021, doi:10.3390/s21113833_

Round 1

Reviewer 1 Report

Major comments: 

The paper studies the muscle synergy extraction methods for moto function evaluation of stroke patients, and a synergy extraction method based on the multivariate curve resolution-alternating least squares scheme is proposed. Extensive experiments with different datasets were conducted to evaluate the performances of the proposed method, and the influences of the sparseness and noise on the extracted synergies. Overall, the paper is well organized, and the work is solid. However, some refinements are still needed in the literature review, analysis and language aspects.

The following changes are recommended for an improved version of the current study:

Minor Comments:

  1. Line 15, the word “spareness” is supposed to be “sparseness”, right? Please double check the typos within the manuscript.
  2. The literature review in the introduction section outlines NMF, PCA, FA, ICA for synergy extraction. Did any of the contemporary studies evaluated common spatial patterns-based methods for synergy extraction? If so, please provide a brief analysis about its merits and demerits in synergy extraction.
  3. How is the similarity between an extracted synergy and the true one synergy defined?
  4. Section 3.5 mentions that the number of synergies has been chosen using the VAF method. Is this the only criteria for synergies number selection? If there’s any other method in literature, how does the VAF compare to it in performance?
  5. How does the synergies extracted from MCR-ALS preserve the biological information of actual synergies?
  6. Font size of ticks and labels values in figure 2 should be increased to improve clarity.
  7. Vocabulary used in the article is relatively plain and could be enhanced.

Reviewer 2 Report

The article presents an important topic for motor function assessment of stroke patients, namely a new muscle synergy extraction method.

I have some minor comments.

Introduction. "So on" should be removed (line 43).

Discussion. The clinical importance of the study should be highlighted.

Reviewer 3 Report

This paper deals with the muscle synergies in healthy subjects and in stroke patients as well as with methods to calculate synergies. The main aim and main findings in the paper are not clearly presented and there are some vague conclusions/statements. though the majority of Methods and materials is sound and appropriate.

Major comments:

  1. Some significant and relevant references related to the muscle synergies are missing throughout the paper. I will name just a few, but please consider adding larger body of knowledge from the literature. Sample references:
    1. Aoi, Shinya, et al. "Neuromusculoskeletal model that walks and runs across a speed range with a few motor control parameter changes based on the muscle synergy hypothesis." Scientific reports 9.1 (2019): 1-13.
    2. Ivanenko, Yuri P., Richard E. Poppele, and Francesco Lacquaniti. "Five basic muscle activation patterns account for muscle activity during human locomotion." The Journal of physiology 556.1 (2004): 267-282.
    3. Pan, Bingyu, et al. "Alterations of muscle synergies during voluntary arm reaching movement in subacute stroke survivors at different levels of impairment." Frontiers in computational neuroscience 12 (2018): 69.
  2. I am wondering whether 40 Hz is too high cut off for high pass filter?
  3. Please, provide a clear and concise aim of the paper.
  4. Do you have a reference for "In addition, the comparative analysis demonstrates that stoke subjects have more synergies and inferior intra-subject consistency in contrast to healthy subjects." as majority of papers deal with the synergistic alteration, but not with higher number of synergies, some papers reported even lesser number of synergies in stroke patients as in Pan, Bingyu, et al. "Alterations of muscle synergies during voluntary arm reaching movement in subacute stroke survivors
     at different levels of impairment." Frontiers in computational neuroscience 12 (2018): 69.
  5. Did you use any clinical measures or tried to correlated synergistic approach with them for results obtained from stroke subjects?

Minor comments:

  1. In abstract - please, add the sample size
  2. Also, how did you measure a performance of the proposed method and why is it important? Please, add this information in abstract
  3. Some grammatical errors are present and minor changes and corrections are needed
  4. It would be much appropriate to present the noise level with the SNR (signal to noise ratio)
  5. Units are written after adding a space next to the number. Please, consider writing "2 s" instead of "2s"
  6. Please, use complete muscle names, i.e., "biceps brachii" instead of just "biceps"
  7. I am not sure what am I looking at Fig. 2? I am not sure what does the color in figures mean?
  8. The paper would benefit from presenting EMG signals in an additional graph

Round 2

Reviewer 1 Report

The authors have successfully addressed all my concerns, and I would like recommed it publication on Sensors in its current form.

Reviewer 3 Report

The manuscript is improved and Authors have correctly answered to all my queries.